ecology, evolution

insect microbiomes, Lepidoptera, host–bacterial associations, metamorphosis

**Author for correspondence:**
Deepa Agashe
e-mail: dagashe@ncbs.res.in

# Disrupting butterfly caterpillar microbiomes does not impact their survival and development

Kruttika Phalnikar, Krushnamegh Kunte and Deepa Agashe

National Centre for Biological Sciences, Tata Institute of Fundamental Research, GKVK Campus, Bellary Road, Bangalore, India

KP, 0000-0002-5582-1055; KK, 0000-0002-3860-6118; DA, 0000-0002-0374-8159

Associations with gut microbes are believed to play crucial roles in the physiology, immune function, development and behaviour of insects. However, microbiome sequencing has recently suggested that butterflies are an anomaly, because their microbiomes do not show strong host- and developmental stage-specific associations. We experimentally manipulated butterfly larval gut microbiota and found that disrupting gut microbes had little influence on larval survival and development. Larvae of the butterflies *Danaus chrysippus* and *Ariadne merione* that fed on chemically sterilized or antibiotic-treated host plant leaves had significantly reduced bacterial loads, and their gut bacterial communities were disrupted substantially. However, neither host species treated this way suffered a significant fitness cost: across multiple experimental blocks, treated and control larvae had similar survival, growth and development. Furthermore, re-introducing microbes from the excreta of control larvae did not improve larval growth and survival. Thus, these butterfly larvae did not appear to rely on specialized gut bacteria for digestion, detoxification, biomass accumulation and metamorphosis. Our experiments thus show that dependence on gut bacteria for growth and survival is not a universal phenomenon across insects. Our findings also caution that strategies which target gut microbiomes may not always succeed in controlling Lepidopteran pests.

## 1. Introduction

The great diversification of insects is thought to have been facilitated by specific and tight associations with gut and other microbes—collectively referred to as the microbiome—that influence host biology in myriad ways [1–3]. For instance, gut microbes of fruit flies influence various aspects of host function such as hormonal signalling [4], metabolism [5], ageing [6], reproduction [7] and behaviour [8]. Many other insects such as termites [9], aphids [10] and honeybees [11] harbour microbes that allow them to survive on highly specialized foods, potentially facilitating diversification into novel dietary niches. This is exemplified by coffee bean borers [12] and oriental fruit flies [13], whose gut microbes detoxify the host diet, allowing survival on otherwise inedible food sources. Thus, gut bacteria can enable hosts to make rapid dietary shifts, with large ecological and evolutionary consequences [14]. These associations can be so important for survival and reproduction that many insects have evolved specific strategies to selectively transmit beneficial microbes across generations [15].

However, exceptions to this pattern of strong insect-microbe interactions are beginning to emerge. In particular, butterflies present a striking contrast because they do not seem to have consistent diet-specific or stage-specific associations with gut bacterial communities. Many wild-caught butterflies harbour similar bacterial communities across the dramatic dietary and developmental transitions that occur during metamorphosis from caterpillars

through pupae to adults [16]. Butterfly caterpillars largely mirror the bacterial communities of their diet, suggesting passive dietary acquisition of gut microbiomes and relatively weak host-imposed selection [16,17]. In Lycaenid butterflies, carnivorous and herbivorous caterpillars harbour similar bacterial communities [18], and diet-induced variation in larval microbiomes does not affect larval growth [19]. These studies indicate that caterpillars may not depend on specific gut bacteria to derive critical nutrition from different dietary resources. Finally, butterfly adults do not bear a reproductive cost of disturbance in gut microbes, even when food is limited [20]. Together, these studies suggest that gut microbiomes of butterflies could be largely transient and may not have functional associations with their hosts. Here, we tested this hypothesis by manipulating microbiomes of butterfly caterpillars (larvae) and measuring various traits related to larval development and survival.

We measured the impact of gut microbes on caterpillars of two butterfly species: *Danaus chrysippus* and *Ariadne merione* (family Nymphalidae) (electronic supplementary material, figure S1). Their caterpillars feed on host plants that use potent anti-herbivory chemical defences. At our study site, *D. chrysippus* caterpillars commonly fed on the locally abundant *Calotropis gigantea* milkweed plant (family Apocynacae), whereas *A. merione* caterpillars usually fed on *Ricinus communis* (castor oil plant) (family Euphorbiaceae) [21]. *Calotropis gigantea* produces white latex in stems and leaves that contain cardiac glycosides which block the activity of the $Na^+/K^+$ pump of herbivores [22,23], rendering the plant poisonous. Similarly, *R. communis* leaves and other tissues contain the alkaloid ricinine that kills insects [24], although its exact mode of action is unknown. Apart from these toxins, consuming plants poses various challenges: leaves are typically difficult to digest, and have low nitrogen content [25].

We experimentally eliminated gut microbes from butterfly larvae using two methods: (i) administering antibiotics through the larval diet, and (ii) chemically sterilizing the larval diet. We conducted these experiments with butterflies, eggs and larval host plants collected from nature, to capture and manipulate naturally occurring microbiomes. We did not find sufficient evidence to reject the null hypothesis that gut bacteria do not affect larval development or survival. Our results, suggesting that the gut microbiomes of butterfly larvae do not aid in digestion and dietary detoxification, run counter to the general trends observed in other insects that feed on toxic food sources [1,26].

## 2. Material and methods

### (a) Insect collection and rearing

We collected *D. chrysippus* males and females on the campus of the National Centre for Biological Sciences (NCBS) (13.0716° N, 77.5794° E). We maintained *D. chrysippus* adults to obtain eggs in cages ($60 \times 30 \times 30$ cm) containing the host plant *C. gigantea*. We kept cages inside a climate-controlled greenhouse maintained at 27–31°C and 60% humidity. In each cage, we kept one female and one to two males (in case the female had not mated in the wild), along with artificial flowers containing artificial nectar solution (Birds Choice no. NP1005, USA). From each female, we obtained 20–80 eggs that were distributed equally into different treatment groups. We obtained *D. chrysippus* eggs in the greenhouse because we found very few eggs on larval host plants in nature. On the other hand, we found relatively large numbers of *A. merione* eggs

on *R. communis* plants around NCBS, which we used directly. We conducted experiments in multiple blocks for each host species. In each block, we split butterfly eggs into specific control and treatment groups as applicable (figure 1; see further details below). For each block, we collected *A. merione* eggs from 5–10 host plants. The number of replicate larvae per treatment varied across blocks, depending on the number of available eggs (see the electronic supplementary material, methods). We housed each larva in the laboratory in a separate plastic container. Every 24–48 h, we supplied larvae with fresh leaves collected from wild plants. We used leaves from three to seven different host plants to include variation across plants and associated microbial communities.

### (b) Antibiotic treatment

We applied two doses of an antibiotic cocktail on leaves presented to *D. chrysippus* and *A. merione* larvae (figure 1). The 'low dose' treatment consisted of ampicillin ($500 \ \mu g \ ml^{-1}$), tetracycline ($50 \ \mu g \ ml^{-1}$) and streptomycin ($100 \ \mu g \ ml^{-1}$) in sterile water, and the 'high dose' treatment contained twice as much of each antibiotic. As a solvent control, we applied sterile double-distilled water (figure 1; electronic supplementary material, methods) and allowed leaf surfaces to dry before feeding larvae. We administered antibiotics with every feeding (every 24–48 h) until pupation.

### (c) Chemical sterilization of diet

We performed experiments with *D. chrysippus* in a laminar hood to minimize contamination by environmental microbes. To eliminate microbes from *C. gigantea* leaves, we dipped them in 70% ethanol for 60 s and 10% bleach for 30 s, followed by three washes with sterile distilled water (figure 1). We dried the leaves and cut them into smaller pieces before feeding the larvae. In the control group, we used untreated leaves. To disentangle the effects of sterilizing agents and microbial elimination, we re-introduced larval gut flora and leaf flora on pre-sterilized leaves using two additional treatments (figure 1). In one treatment, we created a frass (excrement of larval insects) solution by suspending ~500 mg frass from control group larvae in 5 ml of sterile phosphate-buffered saline (PBS). Control group larvae fed on untreated leaves that were expected to harbour the natural microbial community. In the second treatment, we swabbed leaf surfaces of wild *C. gigantea* leaves and suspended the swabs in 5 ml sterile PBS. We painted frass or leaf swab solutions on one side of chemically sterilized leaves, and allowed the leaf surface to dry before feeding larvae (every 24–48 h), until pupation. We did not perform this experiment with *A. merione* because chemically sterilized *R. communis* leaves became limp and lost form permanently when dipped in ethanol and bleach.

### (d) Determining larval gut flora

Detailed protocols of DNA extraction and quantification of the impact of experimental treatments on larval bacterial communities are given in the electronic supplementary material, methods. Briefly, we surface-sterilized and extracted DNA from whole larvae by homogenizing samples with sterile micropestles. We amplified the V3-V4 hypervariable region of the bacterial 16S rRNA gene, and sequenced amplicons on the Illumina MiSeq platform ($300 \times 2$ paired-end reads). We analysed sequenced reads using QIIME [27]. To visualize the differences in microbiomes across larvae with intact (control) versus perturbed (treated) gut flora, we carried out ordination analysis of bacterial communities based on bacterial abundance and composition. We tested whether control and treated samples clustered differently using both principle component analysis and discriminant analysis in R [28]. To validate MiSeq results, we performed quantitative polymerase chain reaction (qPCR) to calculate the abundance of 16S rRNA genes using universal eubacterial primers (forward 5'-TCCTACGGGAGGCAGCAGT-3' and reverse

**Figure 1.** A schematic of manipulative experiments, illustrating the methods used to eliminate gut microbes from host butterfly larvae. (Online version in colour.)

5′-GGACTACCAGGGTATCTAATCCTGTT – 3′) [29] as well as primers specific for dominant bacterial groups (Gammaproteobacteria, Actinobacteria and Firmicutes) relative to an internal control (18S rRNA gene of the host butterfly). For qPCR, we used the same DNA samples that we used for MiSeq analysis and primers specific to the 16S rRNA gene of eubacteria and dominant bacterial groups (see the electronic supplementary material, methods for details).

## (e) Measuring host fitness-related traits

For each host species, we conducted experiments in three to four blocks and measured four to seven fitness proxies (see the electronic supplementary material, tables S5–S7 and methods). We measured larval length (throughout development), larval weight, pupal weight, time taken from hatching until pupation, time taken from pupation until eclosion and the weight of freshly eclosed adults. For some experimental blocks, we also estimated larval digestion efficiency by measuring the gain in larval weight per unit time and per gram of leaf consumed, and the amount of frass produced by larvae per gram of leaf consumed. To visualize broad trends in larval fitness across treatments and blocks, we calculated the average trait value for individuals in each block, and used these data to calculate the mean of means for each trait across all blocks (see the electronic supplementary material, methods). We did not formally analyse the means of means, because with four treatments and two to three blocks in each experiment, we would not have sufficient degrees of freedom to test the impact of treatments.

## (f) Statistical analysis

We used R for all statistical analysis [28]. We analysed data on growth-related traits from all experimental blocks together as well as separately, as follows. For testing the impacts of treatments included in all blocks, we used mixed models to analyse each trait. We specified models with 'treatment' as a fixed effect and 'block' as a random effect in the R package nlme [30], followed by Tukey's HSD for pairwise comparisons across specific treatments using the R package multcomp [31]. We obtained $R^2$ values for mixed

models using the package MuMIn [32]. We report $R_m^2$ ($R^2$ marginal) and $R_c^2$ ($R^2$ conditional), representing the amount of variation explained by models with 'fixed effect' and with 'fixed + random effects' respectively.

Next, we tested whether independent block results corroborated the outcome of the combined analysis across blocks. For growth-related traits, we analysed each experimental block separately using generalized linear models (GLM) followed by Tukey's HSD. For larval survival, we used Fisher's exact test for pairwise comparisons of larval mortality across control and treated groups in each block (for instance, untreated leaves versus sterilized leaves).

Overall, we tested whether removing gut bacteria altered larval growth, resource use, time required for metamorphosis and survival. We could not test the impact of removing bacteria on adult fitness because it was logistically challenging to maintain adults in sterile conditions during mating and oviposition. Furthermore, pupal and adult weights are good predictors of fecundity in the Lepidoptera [33–35], so we could obtain indirect estimates of adult fitness from these measures.

# 3. Results

## (a) Antibiotic treatment and dietary sterilization effectively disrupt larval microbiomes

Our experimental manipulation of gut microbiomes was successful: both antibiotic treatment and dietary sterilization significantly altered bacterial communities (permutational multivariate ANOVA, 10 000 permutations: *D. chrysippus* antibiotic treatment, $p = 0.0045$; *D. chrysippus* dietary sterilization, $p = 0.0047$; *A. merione* antibiotic treatment, $p = 0.0186$; figure 2*a*(i)–*b*(iii); electronic supplementary material, figures S2–S5). Treated caterpillars had lower bacterial loads, reflected in the increased relative abundance of chloroplast and mitochondrial sequence reads (a reduction in the

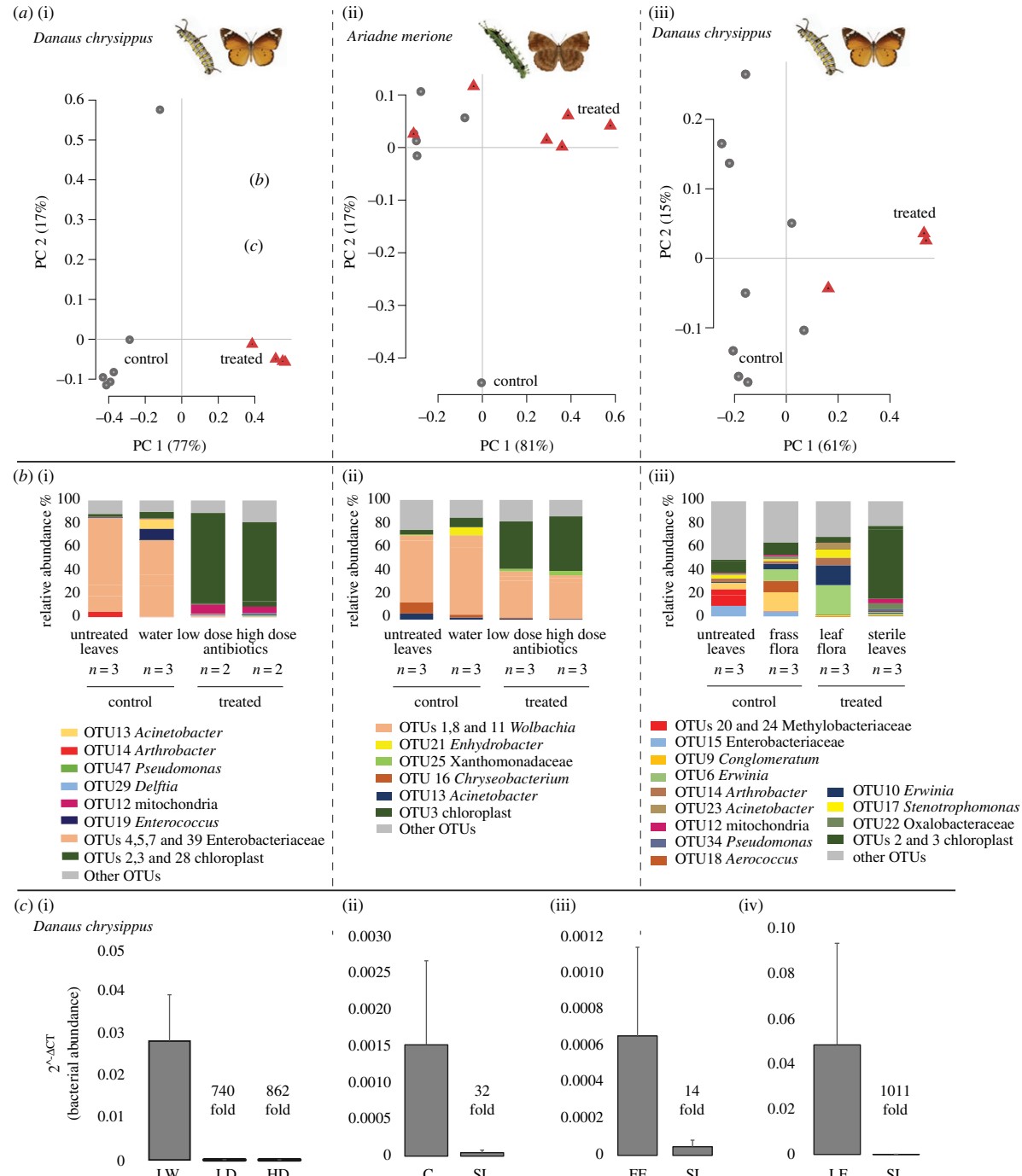

**Figure 2.** Effect of antibiotic treatment and dietary sterilization on bacterial communities of *D. chrysippus* and *A. merione* larvae. (*a*(i)−*a*(iii)) Principle component analysis (PCA) of full bacterial communities from control and treated larvae. Axes show the first two principle components (PC) that explain maximum variation in the data; values in parentheses show the % variation explained by each PC. (*b*(i)−*b*(iii)) Stacked bar plots show the mean relative abundance of the five most abundant bacterial taxa (operational taxonomic units (OTUs)) in each treatment group (see the electronic supplementary material, methods). For microbiome analysis, we used *D. chrysippus* larvae from block 2 (dietary sterilization) and block 3 (antibiotic treatment), and *A. merione* larvae from block 1 (antibiotic treatment) (see the electronic supplementary material, tables S5–S7 and figures S7–S15). (*c*(i)−*c*(iv)) Barplots show the results of quantitative PCR. Magnitude of amplification of the bacterial 16S rRNA gene (using universal bacterial primers) relative to the host 18S rRNA gene (internal control) is calculated as $2^{-\Delta CT}$, where $\Delta CT$ (cycle threshold) $= CT_{target} - CT_{internal\ control}$ ($n = 2$–3 larvae). Error bars represent standard deviation. The mean fold reduction in bacterial abundance in treated versus control samples is indicated ($2^{-\Delta CT}_{control}/2^{-\Delta CT}_{treated}$). See the electronic supplementary material, figure S5 for the results of quantitative PCR for specific dominant bacterial phyla. (Online version in colour.)

number of bacterial 16S gene copies results in greater amplification of leaf-derived chloroplast and leaf- or host-derived mitochondria; electronic supplementary material, figure S6). qPCR further confirmed that bacterial load reduced dramatically after adding antibiotics and sterilizing diet (figure 2*c*(i)–(iv); electronic supplementary material, figure S5).

## (b) Antibiotic treatment does not alter the growth of *Danaus chrysippus* and *Ariadne merione* larvae

Larvae can acquire gut microbes either from their diet or from the egg casing that they often consume [36] and that could harbour specific microbes inoculated by the female. To

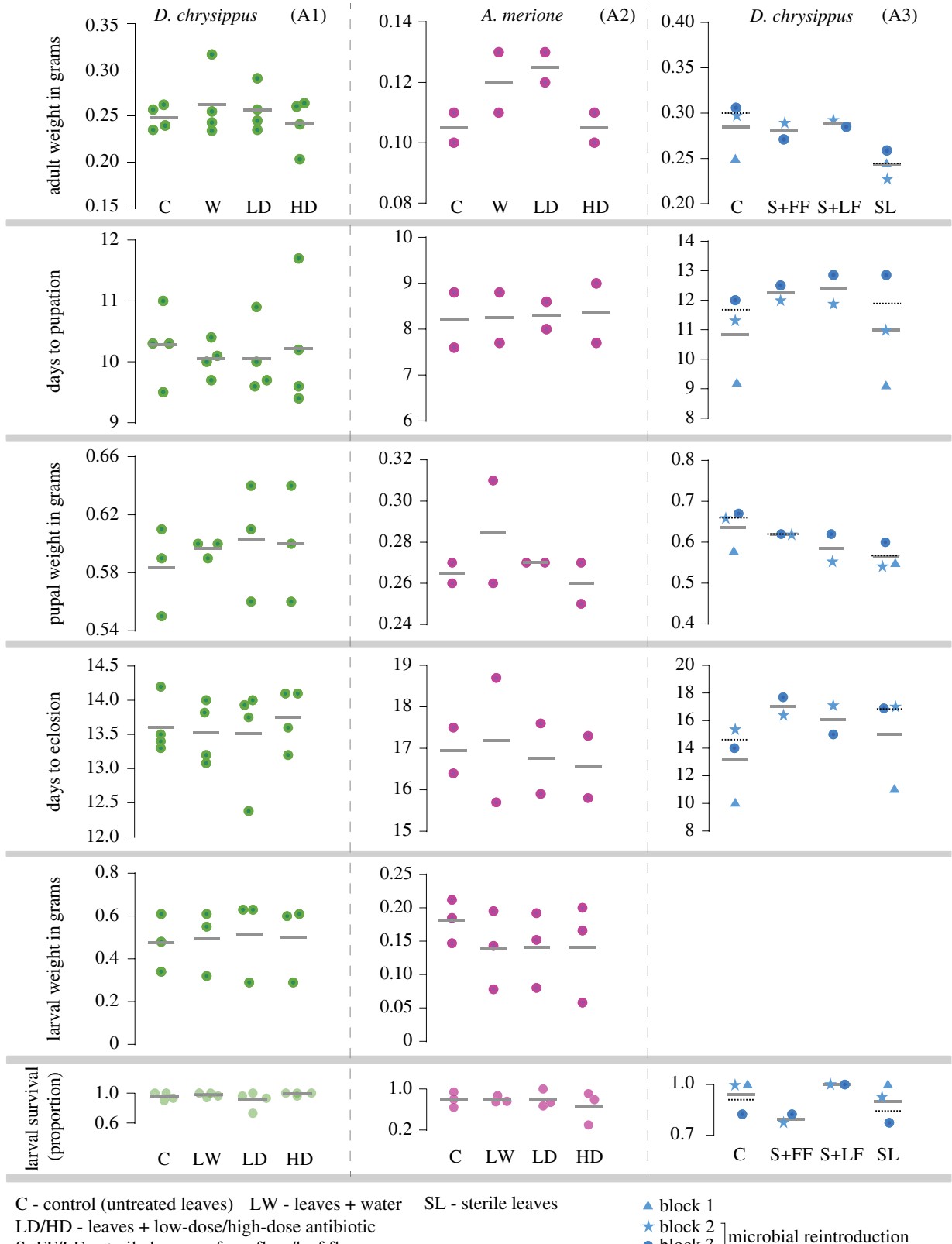

**Figure 3.** Summary of fitness-related traits across treatments. Each point indicates the mean fitness of individuals from a given treatment in each block; grey lines indicate the mean of means across blocks. Black dotted lines in A3 represent the average fitness of blocks 2 and 3 that included microbial reintroduction treatments. See the electronic supplementary material, table S3 for comparisons of block averages across control and treated groups. (Online version in colour.)

eliminate bacteria from these sources, we fed larvae of *D. chrysippus* and *A. merione* with a cocktail of broad-spectrum antibiotics, and tested the impact on their growth and survival. In control groups, we fed larvae either with untreated leaves or with leaves sprayed with water (figure 1). In both butterfly species, the substantial reduction in larval bacterial loads owing to our experimental manipulation (figure 2)

generally did not alter larval growth or survival (figure 3; electronic supplementary material, table S3). A mixed model (all blocks analysed together; model: fitness~treatment, random effect = block) showed that larvae in the sham control group (fed with leaves sprayed with water) were not different from the untreated control group (fed with untreated leaves) or treated groups (fed with leaves +

antibiotics) (electronic supplementary material, tables S1 and S4). The only exception was that *A. merione* pupae treated with a high dose of antibiotics were slightly lighter (by ∼0.02 g) than the sham control; (figure 3; electronic supplementary material, tables S1 and S3). Additionally, in one case the sham control group was significantly different than the untreated group (*A. merione* larval weight; figure 3; electronic supplementary material, table S1). On average, treatments explained 2% of the variation in fitness ($R^2_m$ values in electronic supplementary material, table S1; range 0.4%–6%) whereas treatment and block together explained ∼30% of trait variation ($R^2_c$ values in electronic supplementary material, table S1). Thus, disrupting the larval microbiome via antibiotics did not have strong and significant effects on larval growth and survival.

A separate analysis of data from each block also supported this conclusion (GLM: fitness∼treatment, followed by Tukey's HSD for pairwise comparisons across treatments; electronic supplementary material, figures S7–S12; and tables S5 and S6). In some cases, we observed significant variation across treatments; for instance, pupal weight in *A. merione*, block 2 (electronic supplementary material, table S6). However, this variation was inconsistent across fitness measurements and blocks (electronic supplementary material, figures S7–S12 and tables S5 and S6). Interestingly, larval digestion efficiency also did not vary significantly across control and treated groups for *D. chrysippus* larvae (electronic supplementary material, figures S8–S10; and table S5). The only exception was block 2 (electronic supplementary material table S5 and figure S9), where larvae from the sham control group gained more weight per gram of leaf eaten, compared to larvae fed with low-dose antibiotics. However, this difference was not observed for larvae fed with a high dose of antibiotics. For both butterfly hosts, control and treated larvae had similar survival (electronic supplementary material, table S3, paired *t*-test, $p > 0.05$; electronic supplementary material, table S8, Fisher's exact test, $p > 0.05$), except in *A. merione* block 3 (electronic supplementary material, table S8, Fisher's exact test, $p < 0.02$). In this block, very few larvae fed with a high dose of antibiotics survived to pupation, in contrast to control larvae fed with water-sprayed leaves (electronic supplementary material, figure S12 and table S8). As a result, we could not measure pupal and adult fitness in block 3, and our mixed-model analysis (described above) for *A. merione* did not include data for block 3 (except for larval weight; electronic supplementary material, table S1).

Overall, eliminating gut bacteria using antibiotics did not significantly or consistently alter the measured fitness-related traits in butterfly larvae (electronic supplementary material, tables S1, S3 and S4), which strongly suggests that butterfly larvae do not depend on their gut microbes for growth and survival.

## (c) Dietary sterilization has weak and inconsistent impacts on growth of *Danaus chrysippus* larvae

As an independent method of disrupting the larval bacterial community, we fed *D. chrysippus* caterpillars with surface-sterilized *C. gigantea* leaves. Across three blocks, larvae fed with sterile leaves had significantly lower fitness than control individuals for three out of four fitness measurements (electronic supplementary material, table S2; model: fitness∼treatment, random effect = block); although, sterilization explained very little variation in trait values (average $R^2_m$ across fitness proxies =

0.10; electronic supplementary material, table S2). Thus, in contrast to antibiotic treatments, chemically sterilizing the diet appeared to marginally reduce larval survival and growth.

Given this contrast, we speculated that the chemicals used to sterilize leaves were perhaps toxic for the larvae, potentially confounding the impact of removing microbes. To test this, in the last two blocks of this experiment we included treatments to re-introduce the natural microbiome on chemically sterilized *C. gigantea* leaves (see the electronic supplementary material, methods). These blocks, therefore, had a total of four treatment groups: (i) non-sterile (untreated) leaves, (ii) sterile leaves, (iii) sterile leaves with fecal flora and (iv) sterile leaves with leaf flora (figure 1). If the chemical sterilization were toxic, we expected that re-introducing microbiomes should not rescue fitness, and all treated leaves should be equally detrimental for larvae. By contrast, if low fitness were caused by the lack of microbes, re-introducing microbes should rescue fitness. Across two blocks, re-introducing frass and leaf-surface flora did not fully rescue the fitness of larvae fed with sterilized leaves (figure 3 A3; electronic supplementary material, tables S2 and S4; model: fitness∼treatment, random effect = block; average $R^2_m$ across fitness proxies = 0.13), even though microbial reintroduction was effective (figure 2*a*3,*b*3 and *c*2–*c*4). Only for one fitness measurement (days to pupation), larval fitness varied significantly across untreated leaves and leaves reintroduced with frass flora (electronic supplementary material, table S2; model: fitness∼treatment, random effect = block, $p = 0.001$). Hence, we speculate that sterilizing chemicals were absorbed in the leaf tissue, decreasing larval fitness, with a relatively weak impact of microbial reintroduction (see $R^2_m$ values for model including reintroduction treatments in electronic supplementary material, table S2). This possibility is supported by independent analyses of data from each block, where larval growth or survival were largely unaffected by bacterial elimination or microbial reintroduction (electronic supplementary material, table S7 and figures S13–S15).

Overall, the reduction in bacterial loads and disruption of gut bacterial communities did not impact larval development and survival or adult weight at eclosion (an important predictor of reproductive fitness) in most cases. Together, these results indicate that in our experimental set-up, gut bacteria of butterfly larvae had a negligible impact on larval development and survival.

## 4. Discussion

Associations with gut bacteria impact the fitness of many insects, contributing to the general belief that insect fitness typically depends on their microbiomes [37]. By contrast, our experimental results show that various aspects of butterfly fitness that we measured (growth and survival to adult emergence) are not affected by a substantial disturbance and reduction in their gut microbiomes. Other examples of weak insect-microbiome associations are rare: gut microbes of the eastern spruce budworm do not impact larval growth and survival [38]; stick insect guts are poorly colonized by microbes [39]; dragonfly gut bacterial communities are shaped passively via host diet specialization rather than through selective processes [40]; and fruit fly larvae develop reduced dependence on their gut microbes after evolving on a nutrient-poor diet for several generations [41]. Our results thus provide experimental support for the emerging idea [36] that not all insects have evolved a reliance on specific microbiomes. This is significant not only in

an evolutionary context, but may also have critical implications for insect pest control: a weak dependence on gut microbiome implies that targeting larval gut microbiomes may not always eliminate Lepidopteran and other insect pests.

In conjunction with recent work on butterfly associated bacterial communities (discussed in the Introduction), our experiments show that butterfly larvae may not have established key bacterial mutualisms during their evolution. Moths (including butterflies) also seem to lack a strong association with their gut microbes. For instance, eliminating the gut bacteria of the moth *Manduca sexta* did not impact larval growth, pupal mass and survival [17]. The same study also suggested that Lepidopterans may generally lack strong gut-bacterial associations; and our work provides experimental support for this observation. As suggested previously [17], this lack of host-–bacterial mutualism may arise because changes in butterfly gut morphology and physiology during metamorphosis may prevent the growth and establishment of specific microbiomes. Another possibility is that butterflies evolved a highly efficient and diverse set of digestive enzymes during dietary diversification, allowing larvae to digest various host plants without relying on their gut microbiomes [42]. Finally, butterflies might have evolved microbiome-independent mechanisms to deal with the specific challenges of detoxifying poisonous plants. For instance, milkweed-feeding *D. chrysippus* has evolved cardenolide resistance via mutations in the $Na^+/K^+$ pump [43], potentially weakening any selection favouring detoxification by gut bacteria. However, the relative timescales for the evolution of such host-specific, microbe-independent mechanisms are not clear.

Finally, butterflies may have evolved functional associations with microbes in a different, non-dietary context. For instance, butterfly gut bacteria may play an important role in larval, pupal and adult immune function, as observed in a few other insects [1,44,45]. Our antibiotic treatment may have also eliminated bacterial pathogens, masking the potential role of gut microbes in fighting infections. In addition, as demonstrated earlier in mosquitos [46] and fruit flies [13], gut bacteria could assist in insecticide resistance, which was not tested in this study. More generally, dependence on the microbiome may have evolved in the context of environmental fluctuations (which would be dampened in greenhouse and laboratory experiments such as ours). Another important caveat is that we could not directly test the effect of gut bacteria on adult

fitness. In many insects, gut microbes strongly influence adult foraging [47], fecundity [7], behaviour [48,49] and lifespan [6]. Butterfly adults may derive similar benefits from their microbiomes, although prior work on the microbiomes of wild-caught adult butterflies did not suggest strong host–microbial associations [16,20]. Overall, our results pose an interesting open question: have butterfly caterpillars occupied vastly different dietary niches without recourse to strong gut-bacterial associations, in contrast to the predominant dependence observed in other insects with similarly diverse diets?

In conclusion, the impact of gut microbes on their insect hosts may range along a continuum from strong to weak dependence, to no association. Current literature largely represents only one end of this continuum, whereby gut microbes are thought to strongly affect their hosts. However, the reluctance to publish negative results may have contributed to this general belief, to which our study on wild butterfly larvae presents an interesting contrast. Our work also highlights the need for similar experimental tests of the role of microbiomes in natural populations of other insect hosts with contrasting life histories and diverse niches. We can then begin to understand why some insects critically depend on gut microbes for survival whereas others remain loosely associated.

Data accessibility. Raw data are available (files SRF1–SRF4) in the Dryad Digital Repository: https://dx.doi.org/10.5061/dryad.00000000c [50]. The 16S rRNA MiSeq fastq files are deposited in the European Nucleotide Database (accession ID: PRJEB35305).

Authors' contributions. All authors conceived the project, designed experiments and wrote the manuscript. K.P. performed the experiments. K.P. and D.A. analysed data. K.K. provided infrastructural support and advice on handling butterflies. All authors gave final approval for the manuscript.

Competing interests. We declare we have no competing interests.

Funding. This project was funded by research grants from the National Centre for Biological Sciences to D.A. and K.K., an ICGEB research grant to D.A. (CRP/IND14-01), a Wellcome Trust/DBT India Alliance Fellowship (grant no. IA/I/17/1/503091) to D.A., and a UGC Research Fellowship to K.P.

Acknowledgements. We thank Agashe lab members for their constructive comments on the manuscript; Arun Prakash, Jagrithi Ramnathan, Jagath Vedamurthy and Sanah Imani for laboratory assistance; Kunte lab members for help in the greenhouse and with butterflies; Aparna Agarwal and Rittik Deb for help with 16S MiSeq sequencing and analysis; and Awadhesh Pandit and Tejali Naik from the Next Generation Sequencing (NGS) Facility at NCBS for help with sequencing.

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
