## [Reviewer comments · Proceedings of the Royal Society B: Biological Sciences]

Review History

RSPB-2019-1093.R0 (Original submission)

Review form: Reviewer 1

Recommendation

Major revision is needed (please make suggestions in comments)

Scientific importance: Is the manuscript an original and important contribution to its field?

Good

General interest: Is the paper of sufficient general interest?

Good

Quality of the paper: Is the overall quality of the paper suitable?

Poor

Is the length of the paper justified?

Yes

Should the paper be seen by a specialist statistical reviewer?

No

Do you have any concerns about statistical analyses in this paper? If so, please specify them explicitly in your report.

Yes

It is a condition of publication that authors make their supporting data, code and materials available - either as supplementary material or hosted in an external repository. Please rate, if applicable, the supporting data on the following criteria.

Is it accessible?

No

Is it clear?

N/A

Is it adequate?

N/A

Do you have any ethical concerns with this paper?

No

Comments to the Author

In this study, Phalnikar et al. analyze the effect of microbiota disruption on larval growth and survival in butterflies. Although I find the general research question interesting, both the research field and the study itself is poorly presented in the manuscript.

The host-microbe associations in butterflies seem to parallel what is observed in fruit flies. *Drosophila melanogaster* has been widely used as a model organism to study the effect of microbiota (especially on growth) in the last decade. Although there are over 200 publications studying the effect of microbiota on this insect model, none of the key publications on microbiota transmission and its impact on host physiology (e.g. Storelli et al. 2011, Shin et al. 2011, Blum et al. 2013., Wong et al. 2013, Ridley et al., 2012) is cited or discussed in the manuscript. Moreover, a relatively recent study, Erkosar et al., 2017, (also using the fly model) has shown that in certain conditions hosts can become largely independent from their microbiota for their survival and growth under nutritional stress. This is in direct relation with the subject that is studied here and it has not been discussed.

The main findings of the study are on larval growth and survival phenotypes. Authors presented the data in a table, which is harder to read compared to graphics. Results would have been much more clearly expressed if proper figures were made. Also, it is not clear to me if the authors analyzed the different blocks of experiments together using mixed models, with proper random factors. This should be well described.

The most important control to report concerning the analyses concerns the abundance of the microbiota. To be convincing about the absence of the effect of microbiota depletion on growth and survival, authors must show clearly (by qPCR) that a great decrease in microbiota abundance is observed upon antibiotic treatment. Authors presented this information as a supplement, for different families, but not using universal 16S primers. Ideally, this should be done with universal primers but if for a specific reason (e.g. infection by endosymbionts) this cannot be done, the justification must be made within the text. In any case, since this control is very important, the data must be presented in the main figures.

Overall, although the biological question here is interesting, unfortunately, I do not think that the

quality of the MS is sufficient for publication of the presented study in this journal.

Review form: Reviewer 2

Recommendation

Major revision is needed (please make suggestions in comments)

Scientific importance: Is the manuscript an original and important contribution to its field?

Good

General interest: Is the paper of sufficient general interest?

Excellent

Quality of the paper: Is the overall quality of the paper suitable?

Excellent

Is the length of the paper justified?

Yes

Should the paper be seen by a specialist statistical reviewer?

No

Do you have any concerns about statistical analyses in this paper? If so, please specify them explicitly in your report.

No

It is a condition of publication that authors make their supporting data, code and materials available - either as supplementary material or hosted in an external repository. Please rate, if applicable, the supporting data on the following criteria.

Is it accessible?

No

Is it clear?

Yes

Is it adequate?

No

Do you have any ethical concerns with this paper?

No

Comments to the Author

Summary

Gut microbes can play a number of important roles host biology, but what these roles are (if any) may vary strongly between different taxa. In butterflies and moths, recent sequencing and experimental data indicate that symbionts are not important nutritionally or developmentally, at least for larval stages. The study by Phalnikar et al. adds to this growing body of evidence with an experiment using a comprehensive set of methods to test gut microbial impacts on larval

growth and survival (and other metrics) of two field-collected butterfly species. There are advances over previous experiments on insect gut microbiota, e.g., the use of multiple methods of sterilization (antibiotics and diet chemical sterilization) and, importantly, microbial re-introduction. Sample sizes appear a little low and there is the question of power, when reporting negative results, but the findings seem generally robust to me. The manuscript is well-written and will be of interest to researchers studying insect microbiomes in general and butterflies and moths in particular.

Comments

There are two important caveats that I think are not sufficiently addressed. One, which is partly acknowledged in a limited way (lines 143-147 and 240-243), concerns the possible and unexamined role of gut microbes in adults. Yes, larval traits are major determinants of adult fecundity, but there is also adult nutrition and disease that concerns fitness. This could be acknowledged more clearly. Second, there is a (potentially) important function that was not tested, namely pathogen protection. This is a weakness of all similar experimental studies: by eliminating all microbes with antibiotics or leaf sterilization, potential pathogens are also eliminated. Therefore, if the only job of commensal gut bacteria is to prevent pathogen infection, that will not be evident from these experiments. This should be acknowledged. I do appreciate that the authors are mostly specific in referring to which host traits have been tested for a microbial effect (e.g. lines 23-24).

I suggest that the authors include the qPCR data alongside the compositional data in Figure 2B. Maybe a plot combining all the specific taxa that were quantified. This is more convincing than the compositional data in showing that the treatment eliminated microbes. (But putting both together is the best in my opinion). I'm also unsure as to why the feces/leaf microbial reintroduction data for *D. chrysippus* are not shown in Table 1.

Regarding the occasional negative effects of leaf sterilization on some fitness metrics (line 181-199): This may perhaps be mediated by removal of surface-associated microbes that provide nutrients not as symbionts but in their own biomass, upon being digested. Additionally ethanol sterilization may extract and remove secondary metabolites from the leaf surface or change leaf chemistry in some way. These are mechanisms that do not have to do with the presence/absence of gut-colonizing bacteria.

There is very little information on the measurement and reporting of larval digestive efficiency. I suggest including more detail in the SI methods and also mentioning this (briefly) in the results in the main text. It strengthens the argument that gut bacteria are not involved in development if it is shown that digestion is not affected.

It needs to be explicitly stated how insects were prepared for sequencing. Were the guts dissected? Or was the whole insect homogenized and sequenced? Surface-sterilized? Ideally this information is in the main text methods.

Fig 2B: combine all chloroplast OTUs into one category with one color. I like that chloroplast and mitochondria are shown here but multiple OTUs don't need to be shown. I also suggest grouping all OTUs within the same genus together (e.g., all *Wolbachia* OTUs into one group), so that there are fewer color categories to have to distinguish. Also, given low N of 2-3, consider showing each separately side by side (to visualize variability) rather than means.

SI methods: how many negative controls were extracted and assayed by Qubit? It seems like one, but this should be clearer. Contamination can vary from kit to kit. It is unfortunate that blanks

were not sequenced as there may still be contaminant sequences in the data (e.g. *Delftia*, *Stenotrophomonas* in Table 2).

R code is not yet available in the Supplementary Raw Files. It is important that this is eventually uploaded to Dryad. Also, the sample metadata including barcodes needs to be made publicly available so that the raw sequences can be demultiplexed and re-processed by authors should they want to.

Specific Line Comments

Line 22, change "frass" to feces since this is a non-entomology journal.

Line 33, mosquitoes are not thought to need gut microbes for digestion. See papers by Kerri Coon, Mike Strand and coauthors. This example can just be removed from line 33.

Line 36, ref. 10 does not actually show that gut bacteria detoxify the host diet. (also see the same reference cited on lines 239-240). I would argue that this reference does not report concrete data pertinent to this study.

Lines 214-215: The cited paper by Engel & Moran does not exemplify "the general belief that insect fitness typically depends on their microbiomes", it is actually much more nuanced; for example, it states "... insects vary in the extent to which gut microorganisms are essential or even influential...insects show a wide range in their degree of dependence on gut bacteria". A better reference in which this belief is discussed would be Hammer, Sanders, Fierer FEMS Microbiol Review, <https://doi.org/10.1093/femsle/fnz117>).

Line 216: a more accurate way of phrasing "early growth and survival" would be "growth and survival to adult emergence". This is because no or negligible further growth occurs after metamorphosis so "early" is unnecessary. Also survival between emergence and reproduction, as a function of microbes, could be important in theory but was not tested here.

Line 217-220: it should probably be explicitly mentioned here that caterpillars in general (including butterflies) have been claimed to not require gut microbes (Hammer et al. 2017, reference 14 in the manuscript). Although it is a moth and not a butterfly, this study included similar experimental data on *Manduca sexta*. The current study by Phalnikar et al. is in agreement with these findings, but presents more comprehensive experimental data.

Line 227: change "butterflies" to "butterfly larvae" given that it is still possible that butterfly adults have mutualistic gut bacteria.

Lines 247-248: this point was also made in the aforementioned review paper (<https://doi.org/10.1093/femsle/fnz117>).

Decision letter (RSPB-2019-1093.R0)

17-Jun-2019

Dear Dr Agashe:

I am writing to inform you that your manuscript RSPB-2019-1093 entitled "Disrupting larval

microbiomes does not affect survival and development of host butterflies" has, in its current form, been rejected for publication in Proceedings B.

This action has been taken on the advice of referees, who have recommended that substantial revisions are necessary. With this in mind we would be happy to consider a resubmission, provided the comments of the referees are fully addressed. However please note that this is not a provisional acceptance.

Sincerely,

Professor Gary Carvalho
mailto: proceedingsb@royalsociety.org

Associate Editor
Comments to Author:

This paper reports an experiment on the consequences of microbiota removal on larval life history traits in two species of butterflies. The authors conclude that, in contrast to many other animals, the butterflies seem to suffer no reduction in performance when deprived of the microbiota. This is an interesting finding, and the experiments reported are quite extensive, with two species from different families, multiple experimental blocks and multiple larval traits measured. Thus, it would be an interesting addition to the literature. However, as both reviewers point out, the current version of the paper has several major deficiencies in presentation and statistical analysis; in particular the latter make it unclear to what degree the conclusion is statistically and biologically robust.

Here is the summary of the main points raised by the reviewers plus a few others I find important:

- A non-rejection of a null hypothesis does not constitute sufficient evidence for an absence of an effect. In fact, as the authors must be well aware, an effect can never be formally ruled out. Thus, formally, the statistical conclusion that can be reached here is that the effect is unlikely to be greater than a certain value. This should be quantified by giving confidence intervals on the

effect size (i.e. a difference in the response variables between relevant treatments). This epistemological issue should also be reflected in the language; e.g., the strong statement in l. 17 about "evidence that disrupting microbiota does not affect..." is simply logically wrong. A correct way to state this conclusion would be that any effect is small and unlikely to be biologically relevant; the authors should also discuss why they think the effect of the size that might have been missed would be of little biological relevance.

- The results are presented in a large number of detailed graphs in supplementary figures, and their synthesis in the main text is only reported as a table, which is not very user-friendly. It would be more useful to summarize the results in graphs, e.g. reporting the mean for each treatment x block combination, whereas the table could contain the results of statistical tests and estimates of the effect size and its confidence intervals.

- The statistical analysis is not described in sufficient detail, it is impossible to figure out what has been done exactly. The results of statistical tests are not fully reported, often as $p > 0.05$, without the actual p values, the values of the statistics and df . Table 1 contain estimates of "effect size" (without any measure of confidence), but it is not clear what they are. The text (l. 138) says these are "log odds", which I don't understand given that most traits are not proportion traits; the legend to the table says these are "hedge's d " with no explanation what there are and no reference.

- It seems the authors analyzed each block separately, which reduces power to detect anything and results in a multitude of tests. Data from all blocks should be analyzed together, with block included as a random effect. Furthermore, adjusting P -values for multiple comparison (Tukey test etc.) is conservative from the viewpoint of a conclusion based on rejection of null hypothesis, but anti-conservative / too permissive where non-rejection of null hypothesis is used as a support for the conclusion. Plus, it is not that the authors have no a priori predictions about the various pairwise comparisons. The key comparisons are between the antibiotic and chemical treatments and the sham control. So the Tukey test in this context is misguided.

- In addition to the statistical issue, the robustness of the conclusion is limited by leaving out potential effects on adults and the fact that the design precluded evaluating a potential role of microbiota in protection against pathogens. This should be acknowledged.

- The data on abundance of microbiota are important and so should be included in the main body of the paper.

- As reviewer 1 points out, the authors have ignored a large literature on the effect of microbiota on insects, notably *Drosophila*, including some studies that showed negligible effects of microbiota.

Reviewer(s)' Comments to Author:

Referee: 1

Comments to the Author(s)

In this study, Phalnikar et al. analyze the effect of microbiota disruption on larval growth and survival in butterflies. Although I find the general research question interesting, both the research field and the study itself is poorly presented in the manuscript.

The host-microbe associations in butterflies seem to parallel what is observed in fruit flies. *Drosophila melanogaster* has been widely used as a model organism to study the effect of microbiota (especially on growth) in the last decade. Although there are over 200 publications

studying the effect of microbiota on this insect model, none of the key publications on microbiota transmission and its impact on host physiology (e.g. Storelli et al. 2011, Shin et al. 2011, Blum et al. 2013., Wong et al. 2013, Ridley et al., 2012) is cited or discussed in the manuscript. Moreover, a relatively recent study, Erkosar et al., 2017, (also using the fly model) has shown that in certain conditions hosts can become largely independent from their microbiota for their survival and growth under nutritional stress. This is in direct relation with the subject that is studied here and it has not been discussed.

The main findings of the study are on larval growth and survival phenotypes. Authors presented the data in a table, which is harder to read compared to graphics. Results would have been much more clearly expressed if proper figures were made. Also, it is not clear to me if the authors analyzed the different blocks of experiments together using mixed models, with proper random factors. This should be well described.

The most important control to report concerning the analyses concerns the abundance of the microbiota. To be convincing about the absence of the effect of microbiota depletion on growth and survival, authors must show clearly (by qPCR) that a great decrease in microbiota abundance is observed upon antibiotic treatment. Authors presented this information as a supplement, for different families, but not using universal 16S primers. Ideally, this should be done with universal primers but if for a specific reason (e.g. infection by endosymbionts) this cannot be done, the justification must be made within the text. In any case, since this control is very important, the data must be presented in the main figures.

Overall, although the biological question here is interesting, unfortunately, I do not think that the quality of the MS is sufficient for publication of the presented study in this journal.

Referee: 2

Comments to the Author(s)

Summary

Gut microbes can play a number of important roles host biology, but what these roles are (if any) may vary strongly between different taxa. In butterflies and moths, recent sequencing and experimental data indicate that symbionts are not important nutritionally or developmentally, at least for larval stages. The study by Phalnikar et al. adds to this growing body of evidence with an experiment using a comprehensive set of methods to test gut microbial impacts on larval growth and survival (and other metrics) of two field-collected butterfly species. There are advances over previous experiments on insect gut microbiota, e.g., the use of multiple methods of sterilization (antibiotics and diet chemical sterilization) and, importantly, microbial re-introduction. Sample sizes appear a little low and there is the question of power, when reporting negative results, but the findings seem generally robust to me. The manuscript is well-written and will be of interest to researchers studying insect microbiomes in general and butterflies and moths in particular.

Comments

There are two important caveats that I think are not sufficiently addressed. One, which is partly acknowledged in a limited way (lines 143-147 and 240-243), concerns the possible and unexamined role of gut microbes in adults. Yes, larval traits are major determinants of adult fecundity, but there is also adult nutrition and disease that concerns fitness. This could be acknowledged more clearly. Second, there is a (potentially) important function that was not tested, namely pathogen protection. This is a weakness of all similar experimental studies: by

eliminating all microbes with antibiotics or leaf sterilization, potential pathogens are also eliminated. Therefore, if the only job of commensal gut bacteria is to prevent pathogen infection, that will not be evident from these experiments. This should be acknowledged. I do appreciate that the authors are mostly specific in referring to which host traits have been tested for a microbial effect (e.g. lines 23-24).

I suggest that the authors include the qPCR data alongside the compositional data in Figure 2B. Maybe a plot combining all the specific taxa that were quantified. This is more convincing than the compositional data in showing that the treatment eliminated microbes. (But putting both together is the best in my opinion). I'm also unsure as to why the feces/leaf microbial reintroduction data for *D. chrysippus* are not shown in Table 1.

Regarding the occasional negative effects of leaf sterilization on some fitness metrics (line 181-199): This may perhaps be mediated by removal of surface-associated microbes that provide nutrients not as symbionts but in their own biomass, upon being digested. Additionally ethanol sterilization may extract and remove secondary metabolites from the leaf surface or change leaf chemistry in some way. These are mechanisms that do not have to do with the presence/absence of gut-colonizing bacteria.

There is very little information on the measurement and reporting of larval digestive efficiency. I suggest including more detail in the SI methods and also mentioning this (briefly) in the results in the main text. It strengthens the argument that gut bacteria are not involved in development if it is shown that digestion is not affected.

It needs to be explicitly stated how insects were prepared for sequencing. Were the guts dissected? Or was the whole insect homogenized and sequenced? Surface-sterilized? Ideally this information is in the main text methods.

Fig 2B: combine all chloroplast OTUs into one category with one color. I like that chloroplast and mitochondria are shown here but multiple OTUs don't need to be shown. I also suggest grouping all OTUs within the same genus together (e.g., all *Wolbachia* OTUs into one group), so that there are fewer color categories to have to distinguish. Also, given low N of 2-3, consider showing each separately side by side (to visualize variability) rather than means.

SI methods: how many negative controls were extracted and assayed by Qubit? It seems like one, but this should be clearer. Contamination can vary from kit to kit. It is unfortunate that blanks were not sequenced as there may still be contaminant sequences in the data (e.g. *Delftia*, *Stenotrophomonas* in Table 2).

R code is not yet available in the Supplementary Raw Files. It is important that this is eventually uploaded to Dryad. Also, the sample metadata including barcodes needs to be made publicly available so that the raw sequences can be demultiplexed and re-processed by authors should they want to.

Specific Line Comments

Line 22, change "frass" to feces since this is a non-entomology journal.

Line 33, mosquitoes are not thought to need gut microbes for digestion. See papers by Kerri Coon, Mike Strand and coauthors. This example can just be removed from line 33.

Line 36, ref. 10 does not actually show that gut bacteria detoxify the host diet. (also see the same reference cited on lines 239-240). I would argue that this reference does not report concrete data pertinent to this study.

Lines 214-215: The cited paper by Engel & Moran does not exemplify "the general belief that insect fitness typically depends on their microbiomes", it is actually much more nuanced; for example, it states "... insects vary in the extent to which gut microorganisms are essential or even influential...insects show a wide range in their degree of dependence on gut bacteria". A better reference in which this belief is discussed would be Hammer, Sanders, Fierer FEMS Microbiol Review, <https://doi.org/10.1093/femsle/fnz117>).

Line 216: a more accurate way of phrasing "early growth and survival" would be "growth and survival to adult emergence". This is because no or negligible further growth occurs after metamorphosis so "early" is unnecessary. Also survival between emergence and reproduction, as a function of microbes, could be important in theory but was not tested here.

Line 217-220: it should probably be explicitly mentioned here that caterpillars in general (including butterflies) have been claimed to not require gut microbes (Hammer et al. 2017, reference 14 in the manuscript). Although it is a moth and not a butterfly, this study included similar experimental data on *Manduca sexta*. The current study by Phalnikar et al. is in agreement with these findings, but presents more comprehensive experimental data.

Line 227: change "butterflies" to "butterfly larvae" given that it is still possible that butterfly adults have mutualistic gut bacteria.

Lines 247-248: this point was also made in the aforementioned review paper (<https://doi.org/10.1093/femsle/fnz117>).

Author's Response to Decision Letter for (RSPB-2019-1093.R0)

See Appendix A.

RSPB-2019-2438.R0

Review form: Reviewer 2

Recommendation

Accept with minor revision (please list in comments)

Scientific importance: Is the manuscript an original and important contribution to its field?

Good

General interest: Is the paper of sufficient general interest?

Excellent

Quality of the paper: Is the overall quality of the paper suitable?

Excellent

Is the length of the paper justified?

Yes

Should the paper be seen by a specialist statistical reviewer?

No

Do you have any concerns about statistical analyses in this paper? If so, please specify them explicitly in your report.

No

It is a condition of publication that authors make their supporting data, code and materials available - either as supplementary material or hosted in an external repository. Please rate, if applicable, the supporting data on the following criteria.

Is it accessible?

Yes

Is it clear?

Yes

Is it adequate?

Yes

Do you have any ethical concerns with this paper?

No

Comments to the Author

I am satisfied with the changes the authors have made. The methodology is clearer, and the text is more nuanced, specifically regarding how to discuss negative results, as well as discussing adult-stage microbiomes. Fig. 3 and Fig. 2 C1-C4 do a good job of presenting key results up-front.

The authors should double-check all of their citations as some are mis-numbered or not relevant. To give two examples:

- line 279 ("prior work on the microbiomes of wild-caught adult butterflies..."): the cited Whitaker et al. Front Micro study did not include adult butterflies. Ref. #21 should be #20, to Ravenscraft et al. Mol Ecol.

- lines 255-256 ("..eliminating the gut bacteria of the moth *Manduca sexta*..") cites Chu et al. which is a study on corn rootworms.

This is a comprehensive study that will be a valuable contribution to the field of animal-microbe symbiosis.

Decision letter (RSPB-2019-2438.R0)

07-Nov-2019

Dear Dr Agashe

I am pleased to inform you that your manuscript RSPB-2019-2438 entitled "Disrupting butterfly

caterpillar microbiomes does not impact their survival and development" has been accepted for publication in Proceedings B.

The referee(s) have recommended publication, but also suggest some minor revisions to your manuscript. Therefore, I invite you to respond to the referee(s)' comments and revise your manuscript. Because the schedule for publication is very tight, it is a condition of publication that you submit the revised version of your manuscript within 7 days. If you do not think you will be able to meet this date please let us know.

In order to ensure effective and robust dissemination and appropriate credit to authors the

dataset(s) used should be fully cited. To ensure archived data are available to readers, authors should include a 'data accessibility' section immediately after the acknowledgements section. This should list the database and accession number for all data from the article that has been made publicly available, for instance:

Sincerely,

Professor Gary Carvalho
mailto: proceedingsb@royalsociety.org

Associate Editor
Board Member
Comments to Author:

The authors have addressed the key points and I think the paper will make an interesting contribution to the literature. However, reviewer 2 raised some issues with mis-cited references that the authors should address. Note that the reviewer states that the two issues they identified with references are examples, implying there may be more imprecise references - the authors should thus check all their citations.

Reviewer(s)' Comments to Author:

Referee: 2

Comments to the Author(s).

I am satisfied with the changes the authors have made. The methodology is clearer, and the text is more nuanced, specifically regarding how to discuss negative results, as well as discussing adult-stage microbiomes. Fig. 3 and Fig. 2 C1-C4 do a good job of presenting key results up-front.

The authors should double-check all of their citations as some are mis-numbered or not relevant. To give two examples:

- line 279 ("prior work on the microbiomes of wild-caught adult butterflies..."): the cited Whitaker et al. Front Micro study did not include adult butterflies. Ref. #21 should be #20, to Ravenscraft et al. Mol Ecol.

- lines 255-256 ("..eliminating the gut bacteria of the moth *Manduca sexta*..") cites Chu et al. which is a study on corn rootworms.

This is a comprehensive study that will be a valuable contribution to the field of animal-microbe symbiosis.

Author's Response to Decision Letter for (RSPB-2019-2438.R0)

See Appendix B.

Decision letter (RSPB-2019-2438.R1)

20-Nov-2019

Dear Dr Agashe

I am pleased to inform you that your manuscript entitled "Disrupting butterfly caterpillar microbiomes does not impact their survival and development" has been accepted for publication in Proceedings B.

Open Access

Paper charges

Sincerely,

Proceedings B
mailto: proceedingsb@royalsociety.org

Appendix A

Dear Editor,

Thank you for considering our manuscript for *Proceedings B*. We also thank the reviewers for providing very useful suggestions, which have helped to considerably improve the revision. It took us some time to revise the manuscript because of annual conferences and field work during the summer. However, we have now addressed all the comments by the Associate Editor and reviewers, as detailed below. We hope that the revised manuscript is now acceptable.

Sincerely,

Deepa Agashe
dagashe@ncbs.res.in

Associate Editor

Comments to Author:

This paper reports an experiment on the consequences of microbiota removal on larval life history traits in two species of butterflies. The authors conclude that, in contrast to many other animals, the butterflies seem to suffer no reduction in performance when deprived of the microbiota. This is an interesting finding, and the experiments reported are quite extensive, with two species from different families, multiple experimental blocks and multiple larval traits measured. Thus, it would be an interesting addition to the literature. However, as both reviewers point out, the current version of the paper has several major deficiencies in presentation and statistical analysis; in particular the latter make it unclear to what degree the conclusion is statistically and biologically robust.

Here is the summary of the main points raised by the reviewers plus a few others I find important:

- 1) A non-rejection of a null hypothesis does not constitute sufficient evidence for an absence of an effect. In fact, as the authors must be well aware, an effect can never be formally ruled out. Thus, formally, the statistical conclusion that can be reached here is that the effect is unlikely to be greater than a certain value. This should be quantified by giving confidence intervals on the effect size (i.e. a difference in the response variables between relevant treatments). This epistemological issue should also be reflected in the language; e.g., the strong statement in l. 17 about "evidence that disrupting microbiota does not affect..." is simply logically wrong. A correct way to state this conclusion would be that any effect is small and unlikely to be biologically relevant; the authors should also discuss why they think the effect of the size that might have been missed would be of little biological relevance."

Response: We have modified the language in the manuscript, as suggested (for instance, page 3, line 60 and page 8, line 232). We have also included effect sizes and confidence intervals for our main analysis (tables 1, S1 and S2).

- 2) "The results are presented in a large number of detailed graphs in supplementary figures, and their synthesis in the main text is only reported as a table, which is not very user-friendly. It would be more useful to summarize the results in graphs, e.g. reporting the mean for each treatment x block combination, whereas the table could contain the results of statistical tests and estimates of the effect size and its confidence intervals."

Response: We have now added a figure summarizing key results across blocks (Fig 3) and included effect sizes and confidence intervals for all statistical analysis (tables 1, 2, S1 and S2).

- 3) “The statistical analysis is not described in sufficient detail, it is impossible to figure out what has been done exactly. The results of statistical tests are not fully reported, often as $p > 0.05$, without the actual p values, the values of the statistics and df. Table 1 contain estimates of "effect size" (without any measure of confidence), but it is not clear what they are. The text (l. 138) says these are "log odds", which I don't understand given that most traits are not proportion traits; the legend to the table says these are "hedge's d" with no explanation what there are and no reference.”

Response: We had actually reported all statistical details in the supplementary information. We have now elaborated on our statistical methods (page 5, line 133 in the main text and page 18, line 205 in SI methods). In Tables 1 and 2 of the main manuscript, we also report exact p values, effect sizes, degrees of freedom and confidence intervals for all fitness comparisons. We have not mentioned exact p values in the main text because a single statement in the text often sums up results of multiple fitness traits; hence, we refer to the appropriate tables with detailed statistics. We have also clarified the meaning and interpretation of Hedge's g , and why we are using it, although now we have moved this explanation to supplementary information. We re-analyzed data using linear mixed models (as suggested in the next point), and we discuss our results based on R^2 values. We no longer use log odds in our analysis.

- 4) “It seems the authors analyzed each block separately, which reduces power to detect anything and results in a multitude of tests. Data from all blocks should be analyzed together, with block included as a random effect. Furthermore, adjusting P -values for multiple comparison (Tukey test etc.) is conservative from the viewpoint of a conclusion based on rejection of null hypothesis, but anti-conservative / too permissive where non-rejection of null hypothesis is used as a support for the conclusion. Plus, it is not that the authors have no a priori predictions about the various pairwise comparisons. The key comparisons are between the antibiotic and chemical treatments and the sham control. So the Tukey test in this context is misguided.”

Response: Previously, we chose to analyse each block separately because in the dietary sterilization regime all blocks did not have identical treatments. However, as suggested, we have now performed a mixed-model analysis of all experimental blocks together, with “block” included as a random effect (see table 1). For this analysis, we only considered treatments that were shared across all blocks. To determine the impacts of block-specific treatments, we have continued to use block-specific analyses. We think that Tukey's HSD is necessary in our analysis since all the pairwise comparisons are informative. Hence, we have retained the results obtained using Tukey's HSD, but have moved these to the supplementary information for interested readers (see table S2).

- 5) “In addition to the statistical issue, the robustness of the conclusion is limited by leaving out potential effects on adults and the fact that the design precluded evaluating a potential role of microbiota in protection against pathogens. This should be acknowledged.”

Response: We had acknowledged both these issues, which are retained in the revision (page 5, line 144, page 8, lines 236-243 in the original submission; page 9, lines 268-279 in the revised manuscript).

- 6) “The data on abundance of microbiota are important and so should be included in the main body of the paper.”

Response: We have now included these data in Fig 2, panels C1-C4.

- 7) “As reviewer 1 points out, the authors have ignored a large literature on the effect of microbiota on insects, notably *Drosophila*, including some studies that showed negligible effects of microbiota.”

Response: We recognize that the literature on *Drosophila* microbiota is quite extensive, but we could not possibly do justice to that literature and provide enough citations in this paper. Instead, we have tried to summarize broad patterns of gut microbiome effects across diverse groups of insects, including a few examples of *Drosophila* (page 2, lines 19-29). We have also discussed one recent study on wild *Drosophila* populations that reported negative results (page 8, line 245). The remaining literature may be traced from the review papers that we have cited (page 2, lines 19-21).

Referee 1

“In this study, Phalnikar et al. analyze the effect of microbiota disruption on larval growth and survival in butterflies. Although I find the general research question interesting, both the research field and the study itself is poorly presented in the manuscript.

- 1) The host-microbe associations in butterflies seem to parallel what is observed in fruit flies. *Drosophila melanogaster* has been widely used as a model organism to study the effect of microbiota (especially on growth) in the last decade. Although there are over 200 publications studying the effect of microbiota on this insect model, none of the key publications on microbiota transmission and its impact on host physiology (e.g. Storelli et al. 2011, Shin et al. 2011, Blum et al. 2013., Wong et al. 2013, Ridley et al., 2012) is cited or discussed in the manuscript. Moreover, a relatively recent study, Erkosar et al., 2017, (also using the fly model) has shown that in certain conditions hosts can become largely independent from their microbiota for their survival and growth under nutritional stress. This is in direct relation with the subject that is studied here and it has not been discussed.”

Response: We have now included an example with *Drosophila* in the introduction. Since we want to discuss broader patterns observed across diverse insects, we cannot fully discuss the vast literature on flies. Host-microbe associations in flies are actually not parallel to our observations with butterflies: most prior work shows that *Drosophila* species have a strong association with their gut microbes and depend on them for survival, development and reproduction. However, we thank the reviewer for pointing out the *Eskosar et al* paper, which we now cite and discuss (page 8, lines 245-246).

- 2) “The main findings of the study are on larval growth and survival phenotypes. Authors presented the data in a table, which is harder to read compared to graphics. Results would have been much more clearly expressed if proper figures were made. Also, it is not clear to me if the authors analyzed the different blocks of experiments together using mixed models, with proper random factors. This should be well described.”

Response: We have added a new figure summarizing key results (Fig 3), and clarified statistical analyses in the Methods section (page 5, line 133; please also see our response to the Editor’s comment above).

- 3) “The most important control to report concerning the analyses concerns the abundance of the microbiota. To be convincing about the absence of the effect of microbiota depletion on growth and survival, authors must show clearly (by qPCR) that a great decrease in microbiota abundance is observed upon antibiotic treatment. Authors presented this information as a supplement, for different families, but not using universal 16S primers. Ideally, this should be done with universal primers but if for a specific reason (e.g. infection by endosymbionts) this cannot be done, the justification must be made within the text. In any case, since this control is very important, the data must be presented in the main figures.”

Response: We have now included qPCR data in a main figure (Fig 2). As suggested, we conducted qPCR analysis using universal eubacterial primers (Fig 2, panels C1-C4; also see SI methods, page 17, line 181). qPCR results for individual bacterial phyla are shown in Fig S5.

Referee: 2

Comments to the Author(s)

Summary: Gut microbes can play a number of important roles host biology, but what these roles are (if any) may vary strongly between different taxa. In butterflies and moths, recent sequencing and experimental data indicate that symbionts are not important nutritionally or developmentally, at least for larval stages. The study by Phalnikar et al. adds to this growing body of evidence with an experiment using a comprehensive set of methods to test gut microbial impacts on larval growth and survival (and other metrics) of two field-collected butterfly species. There are advances over previous experiments on insect gut microbiota, e.g., the use of multiple methods of sterilization (antibiotics and diet chemical sterilization) and, importantly, microbial re-introduction. Sample sizes appear a little low and there is the question of power, when reporting negative results, but the findings seem generally robust to me. The manuscript is well-written and will be of interest to researchers studying insect microbiomes in general and butterflies and moths in particular.

Comments

- 1) There are two important caveats that I think are not sufficiently addressed. One, which is partly acknowledged in a limited way (lines 143-147 and 240-243), concerns the possible and unexamined role of gut microbes in adults. Yes, larval traits are major determinants of adult fecundity, but there is also adult nutrition and disease that concerns fitness. This could be acknowledged more clearly. Second, there is a (potentially) important function that was not tested, namely pathogen protection. This is a weakness of all similar experimental studies: by eliminating all microbes with antibiotics or leaf sterilization, potential pathogens are also eliminated. Therefore, if the only job of commensal gut bacteria is to prevent pathogen infection, that will not be evident from these experiments. This should be acknowledged. I do appreciate that the authors are mostly specific in referring to which host traits have been tested for a microbial effect (e.g. lines 23-24)."

Response: Both the caveats are now acknowledged more clearly in the revised manuscript (page 9, lines 268-279)

- 2) "I suggest that the authors include the qPCR data alongside the compositional data in Figure 2B. Maybe a plot combining all the specific taxa that were quantified. This is more convincing than the compositional data in showing that the treatment eliminated microbes. (But putting both together is the best in my opinion). I'm also unsure as to why the feces/leaf microbial reintroduction data for *D. chrysippus* are not shown in Table 1."

Response: We have now included qPCR data in the main figure (Fig 2), and microbial reintroduction data in table 2, table S1 and S2.

- 3) "Regarding the occasional negative effects of leaf sterilization on some fitness metrics (line 181-199): This may perhaps be mediated by removal of surface-associated microbes that provide nutrients not as symbionts but in their own biomass, upon being digested. Additionally ethanol sterilization may extract and remove secondary metabolites from the leaf surface or change leaf chemistry in some way. These are mechanisms that do not have to do with the presence/absence of gut-colonizing bacteria."

Response: We appreciate these interesting points, and have now discussed these possibilities (pages 7-8, line 211-228).

- 4) “There is very little information on the measurement and reporting of larval digestive efficiency. I suggest including more detail in the SI methods and also mentioning this (briefly) in the results in the main text. It strengthens the argument that gut bacteria are not involved in development if it is shown that digestion is not affected.”

Response: We have added more information as suggested (main methods, page 5, line 125; SI, page 18 line 198).

- 5) “It needs to be explicitly stated how insects were prepared for sequencing. Were the guts dissected? Or was the whole insect homogenized and sequenced? Surface-sterilized? Ideally this information is in the main text methods.”

Response: We have now added this information in the main text and supplementary methods (main methods, page 4, line 107; SI methods, page 17, lines 147-150).

- 6) “Fig 2B: combine all chloroplast OTUs into one category with one color. I like that chloroplast and mitochondria are shown here but multiple OTUs don't need to be shown. I also suggest grouping all OTUs within the same genus together (e.g., all *Wolbachia* OTUs into one group), so that there are fewer color categories to have to distinguish. Also, given low N of 2-3, consider showing each separately side by side (to visualize variability) rather than means.”

Response: We have now combined OTUs as suggested (Fig 2, panels B1-B3). We have also included a new figure (Fig S2) showing OTUs of individual larvae.

- 7) “SI methods: how many negative controls were extracted and assayed by Qubit? It seems like one, but this should be clearer. Contamination can vary from kit to kit. It is unfortunate that blanks were not sequenced as there may still be contaminant sequences in the data (e.g. *Delftia*, *Stenotrophomonas* in Table 2).”

Response: Only one negative control was used, which is now clearly stated in SI methods (page 17, line 151). We agree that sequencing the negative controls would have helped us to identify the contaminating bacterial OTUs. However, since the concentration of the negative control was below detection for a highly sensitive technique like Qubit, we suspected that contaminating OTUs would probably contribute very few reads to the entire dataset. Hence, we did not sequence the negative control.

- 8) “R code is not yet available in the Supplementary Raw Files. It is important that this is eventually uploaded to Dryad. Also, the sample metadata including barcodes needs to be made publicly available so that the raw sequences can be demultiplexed and re-processed by authors should they want to.”

Response: We will certainly make the R codes and sample metadata available after the manuscript is accepted.

Specific Line Comments

9) “Line 22, change "frass" to feces since this is a non-entomology journal.”

Response: This is a standard term, so we would like to retain it. At the first mention of frass, we have now included ‘excrement of larval insects’ as an explanation in parentheses (page 4, line 97).

10) “Line 33, mosquitoes are not thought to need gut microbes for digestion. See papers by Kerri Coon, Mike Strand and coauthors. This example can just be removed from line 33.”

Response: We have now removed this reference.

11) “Line 36, ref. 10 does not actually show that gut bacteria detoxify the host diet. (also see the same reference cited on lines 239-240). I would argue that this reference does not report concrete data pertinent to this study.”

Response: We have removed this reference in the revision.

12) “Lines 214-215: The cited paper by Engel & Moran does not exemplify "the general belief that insect fitness typically depends on their microbiomes", it is actually much more nuanced; for example, it states "... insects vary in the extent to which gut microorganisms are essential or even influential...insects show a wide range in their degree of dependence on gut bacteria". A better reference in which this belief is discussed would be Hammer, Sanders, Fierer FEMS Microbiol Review, <https://doi.org/10.1093/femsle/fnz117>.”

Response: We have changed the reference as suggested (page 8, lines 238- 239, reference number 36).

13) “Line 216: a more accurate way of phrasing "early growth and survival" would be "growth and survival to adult emergence". This is because no or negligible further growth occurs after metamorphosis so "early" is unnecessary. Also survival between emergence and reproduction, as a function of microbes, could be important in theory but was not tested here.”

Response: We have made this correction (page 8, line 240)

14) “Line 217-220: it should probably be explicitly mentioned here that caterpillars in general (including butterflies) have been claimed to not require gut microbes (Hammer et al. 2017, reference 14 in the manuscript). Although it is a moth and not a butterfly, this study included similar experimental data on *Manduca sexta*. The current study by Phalnikar et al. is in agreement with these findings, but presents more comprehensive experimental data.”

Response: We have now added this point in the discussion (page 9, line 254-257).

15) “Line 227: change "butterflies" to "butterfly larvae" given that it is still possible that butterfly adults have mutualistic gut bacteria.”

Response: We have made this change throughout the manuscript.

16) “Lines 247-248: this point was also made in the aforementioned review paper (<https://doi.org/10.1093/femsle/fnz117>).”

Response: We acknowledge that both this review paper and our manuscript (posted on Biorxiv earlier) made this point in parallel. We have therefore also cited the review (reference number 36).

Appendix B

RSPB-2019-2438

Disrupting butterfly caterpillar microbiomes does not impact their survival and development

Dear Editor,

Thank you for accepting our manuscript for *Proceedings B*. We also thank the reviewers for pointing out the errors in citations.

Sincerely,

Deepa Agashe
dagashe@ncbs.res.in

Referee: 2

Comments to the Author(s).

The authors should double-check all of their citations as some are mis-numbered or not relevant. To give two examples:

- line 279 ("prior work on the microbiomes of wild-caught adult butterflies..."): the cited Whitaker et al. Front Micro study did not include adult butterflies. Ref. #21 should be #20, to Ravenscraft et al. Mol Ecol.
- lines 255-256 ("..eliminating the gut bacteria of the moth *Manduca sexta*..") cites Chu et al. which is a study on corn rootworms.

Response: We have corrected the citations as suggested.

- We have corrected both the errors pointed out by the reviewer
- In the introduction we have made a minor alteration in a sentence, to make it more relevant for the references we have cited (page 2, paragraph1, line 23).
- Reference number 4 is corrected from "Storelli et al., 2018" to "Storelli et al., 2011".
- We have added a new reference (number 29, page 4, line 117), which was missing earlier
- We have added reference number 49 to provide more support for the related statement (page 9, paragraph 2, line 278).

In addition, we have included a new funding source which was missing earlier - A Wellcome Trust/DBT India Alliance Fellowship (grant number IA/I/17/1/503091) to Deepa Agashe.